# Preparation of Polyacrylate Hollow Microspheres via Facile Spray Drying

**Pingxu Chen** [1,*], **Nanbiao Ye** [1], **Chaoxiong He** [1], **Lei Tang** [1], **Shuliang Li** [2,3], **Luyi Sun** [3,*] and **Yuntao Li** [2]

[1] National Engineering Laboratory of Plastics Modification and Processing, and Research and Development Center, Kingfa Science and Technology Company, Ltd., Guangzhou 510663, China; yenanbiao@kingfa.com.cn (N.Y.); hechaoxiong@kingfa.com.cn (C.H.); tanglei@kingfa.com.cn (L.T.)

[2] College of Materials Science and Engineering, Southwest Petroleum University, Chengdu 610050, China; shuliang.li@stu.swpu.edu.cn (S.L.); yuntaoli@swpu.edu.cn (Y.L.)

[3] Polymer Program, Institute of Materials Science and Department of Chemical & Biomolecular Engineering, University of Connecticut, Storrs, Connecticut, CT 06269, USA

[*] Correspondence: pinger@kingfa.com.cn (P.C.); luyi.sun@uconn.edu (L.S.)

**Abstract:** Polyacrylate microspheres with a hollow structure were prepared by a facile spray drying method. The effects of spray drying process parameters, including inlet temperature, atomizer rotational speed, and feed speed, on the particle size, bulk density, and morphology of the resultant polyacrylate hollow microspheres were investigated and discussed. The mechanism for the formation of the polyacrylate hollow microspheres was proposed. This facile and scalable method for preparing hollow polymer microspheres is expected to be valuable to prepare various polymer hollow structures for widespread application.

**Keywords:** polyacrylate; hollow microspheres; spray drying

## 1. Introduction

Polymer hollow microspheres have drawn major attention because of their large specific surface area, relative low density, and high encapsulation capability [1–4]. As a result, they have found a wide range of applications, including drug delivery, catalysis, and coatings [5–7]. Specifically, hollow microspheres can be used as drug carriers, improving the flowability and packability when compared with the raw crystals of drugs [8]. Single-hole hollow polymer microspheres with specific high-capacity uptake of target species may also provide new opportunities in the capsulation of drugs [9]. Silica/polymer hollow functional polymer microspheres prepared by reversible addition–fragmentation chain transfer (RAFT) polymerization were applied as a reservoir for nitric oxide (NO) [10]. Moreover, conductive polymer hollow microspheres with high specific surface areas and outstanding electrical properties exhibited superb microwave absorption performance [11]. Therefore, it is meaningful to develop a facile and scalable method to prepare hollow microspheres for various applications.

Polymer hollow microspheres are mainly prepared via three approaches: self-assembly, template assisted synthesis, and emulsion polymerization [12–15]. Self-assembly and template assisted synthesis are demanding on reaction system and processing procedures, while emulsion polymerization is more industry-friendly and scalable. In addition, this emulsion polymerization method can effectively control sphere size, structure, and composition. Therefore, emulsion polymerization is an ideal method to prepare polymer hollow microspheres, especially at a large scale [16,17]. Kobayashi et al. prepared hollow polystyrene particles by seeded emulsion polymerization [18], which has many advantages,

including being less dependent on the type of base polymer used, more friendly to environment, and more easy to scale up. However, further water absorption by the particles may occur during the polymerization, thus making it challenging to control the hollow structure.

In recent years, preparation of polymer microspheres by spray drying has attracted attention, in which an atomizer is employed to make small droplets with subsequent flash evaporation by hot air to obtain microspheres. Compared with other methods, spray drying has the advantages of a facile process, high controllability, and ease of commercialization [19,20]. However, preparing hollow microspheres with controllable size and morphology by spray drying remains a challenge. We aim to develop a facile and scalable method to prepare polymer hollow microspheres by combining emulsion polymerization and spray drying.

## 2. Materials and Methods

A high-speed centrifugal spray dryer (model LPG-8, Changzhou Lima Drying Engineering Co., Ltd., Changzhou, China) was used. A conventional polyacrylate emulsion was used as the model emulsion for this project to prepare polyacrylate hollow spheres, while other polymer emulsions should work similarly effectively. First, a polyacrylate emulsion with a solid content of 33 wt % and latex particle size of 30–50 nm was prepared by using methyl methacrylate acrylate (MMA)/butyl acrylate (BA)/acrylic acid (AA) (85/15/3 in mass ratio) as comonomers (MMA, BA, and AA were all industrial grade and obtained from Guangzhou Guanglin Chemical Co., Ltd., Guangzhou, China) and using ammonium sulfate allyloxy nonylphenoxy poly(ethyleneoxy)(10) ether (DNS-86, industrial grade, Guangzhou Shuangjian Trade Co., Ltd., Guangzhou, China) as emulsifier via semi-continuous emulsion polymerization [21]. The glass transition temperature ($T_g$) of the synthesized polyacrylate was characterized to be 83 °C by differential scanning calorimetry (DSC) at a heating rate of 10 °C/min.

The synthesized polyacrylate emulsion was filtered with a 200 mesh filter and transferred into a centrifugal spray drying tower (Figure 1) using a peristaltic pump (model BT300, Changzhou Purui Fluid Technology Co., Ltd., Changzhou, China). The emulsion was subsequently converted into small droplets by a high-speed atomizer (model DPG-5, Xiangsu Xinglun Electromechanical Equipment Co., Ltd., Taizhou, China) because of the atomization effect. The formed small droplets were in full contact with the hot air in the tower body, and were dried into microspheres after being heat-exchanged along their transportation path, and then separated by a cyclone separator. The microspheres were finally collected in a receiving tank and filtered through a 100-mesh filter and then sealed. The entire process is briefly shown in Figure 1. The resultant microspheres were characterized by scanning electron microscopy (SEM) and laser diffraction. For SEM, the polyacrylate microspheres were coated with a thin gold layer before being imaged using a HITACHI S-3400N scanning electron microscope (HITACHI, Hitachinaka, Japan) operated at a working voltage of 20 kV. For laser diffraction, the microspheres were dispersed in deionized water to prepare a dilute suspension (0.1–1.0 wt %) with the assistance of ultrasonication, and subsequently characterized on a Malvern Laser Particle Sizer (Mastersizer 3000, Malvern, UK). Three specimens of each sample were characterized, and the average results were recorded.

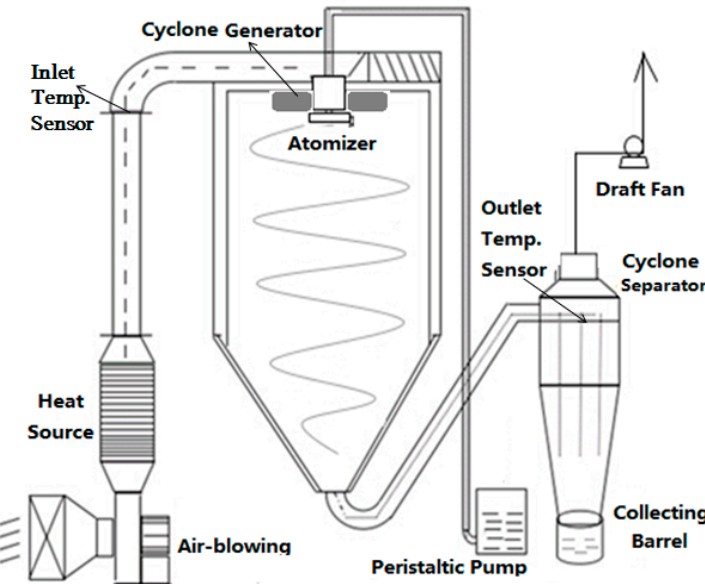

**Figure 1.** Schematic of a high-speed centrifugal spray dryer.

## 3. Results and Discussion

During the entire process, three key parameters affect the drying result and therefore the morphology of the formed polyacrylate hollow microspheres: (1) temperature, including inlet air temperature (i.e., the temperature of the hot air enters the tower) and outlet air temperature (i.e., the temperature of the cyclone separator above the receiver); (2) size of the droplets, which is mainly decided by the speed of the atomizer; and (3) feed rate of the emulsion.

Properly tuning the inlet and the outlet air temperatures is particularly critical to the formation of the polyacrylate hollow microspheres. These temperatures not only affect the water evaporation rate, moisture content, and drying efficiency, but also directly determine the microsphere nucleation mode, thereby affecting the microsphere size and morphology. The effect of inlet and outlet temperatures on the microsphere bulk density is illustrated in Table 1. At a 28,000 rpm atomizer rotational speed and an emulsion feed rate of 160 g/min, the bulk density of the hollow microspheres decreased with an increasing inlet/outlet temperature, especially when the inlet air temperature was at 200–220 °C. As shown in Figure 2, inlet air temperature has an effect on the size, size distribution, and morphology of the prepared polyacrylate microspheres. The microsphere size increased very marginally when the inlet temperature was increased from 140 to 200 °C. However, a slight size increase occurred in the range of 200–220 °C. For example, the D(50) of the microspheres increased from 34 to 43 μm when the inlet temperature was increased from 200 to 220 °C. The SEM images in Figure 2 show the morphology and surface structure of the microspheres prepared at various temperatures. One can observe that most of particles show a spherical morphology and a large portion of them are hollow, as shown in Figure 2a. Meanwhile, a small amount of broken fragments of microspheres are observed in Figure 2a–c, which is consistent with the literature [22].

**Table 1.** Effect of inlet/outlet temperature on the bulk density and dimensions of the microspheres.

| Inlet Temperature/°C | Outlet Temperature/°C | bulk Density/(g/cm$^3$) | D(10)/µm | D(50)/µm | D(90)/µm |
|---|---|---|---|---|---|
| 140 | 61 | 0.52 | 16 | 31 | 58 |
| 160 | 66 | 0.50 | 18 | 33 | 58 |
| 180 | 73 | 0.49 | 18 | 33 | 57 |
| 200 | 84 | 0.44 | 18 | 34 | 60 |
| 220 | 98 | 0.37 | 24 | 43 | 74 |

D(10), D(50), and D(90) particle sizes are the equivalent diameters of the largest particles with a cumulative volume distribution of 10%, 50%, and 90% in the distribution curve, respectively.

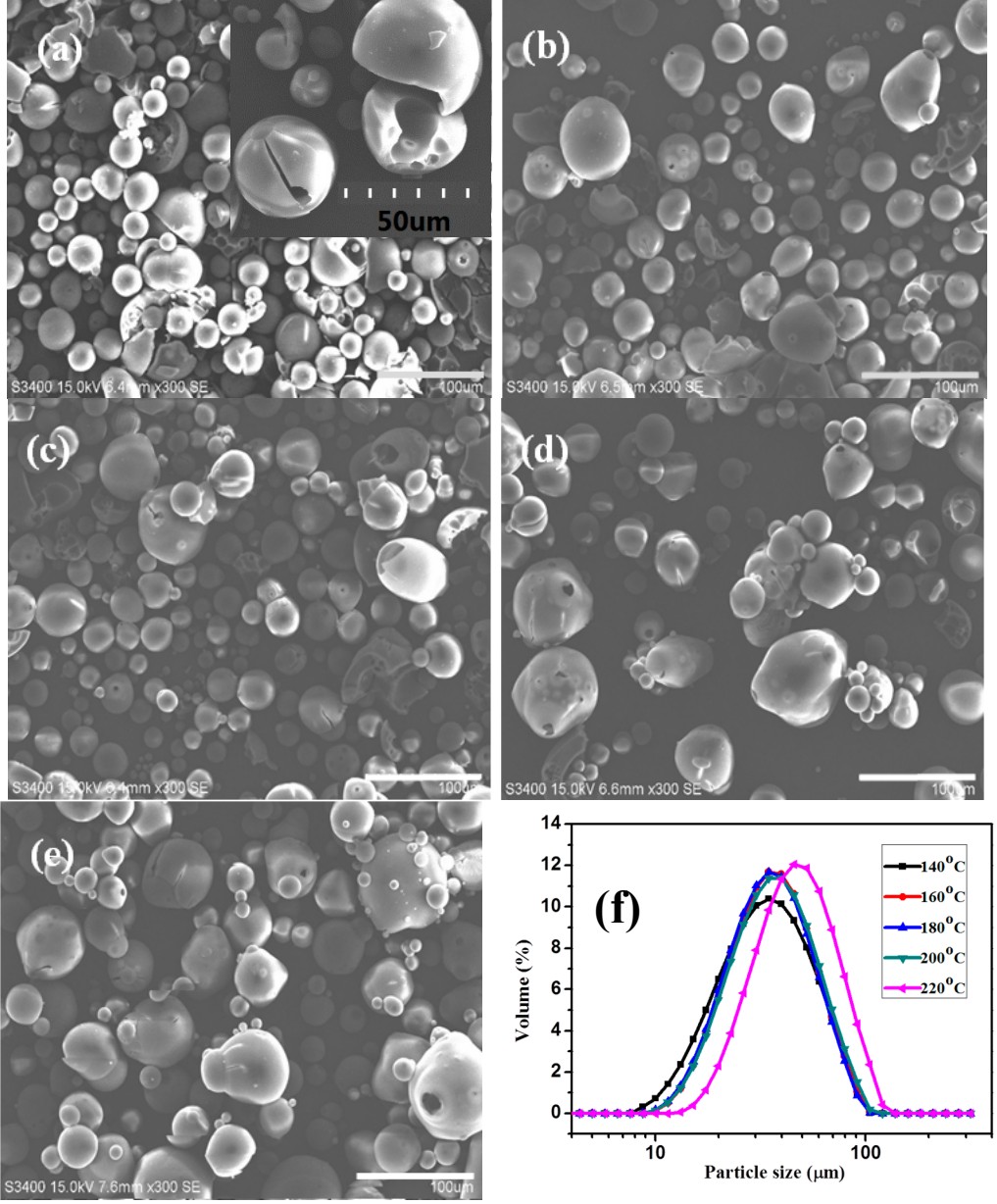

**Figure 2.** SEM images of the polyacrylate hollow microspheres prepared at different inlet temperatures: (**a**) 140 °C; (**b**) 160 °C, (**c**) 180 °C, (**d**) 200 °C, and (**e**) 220 °C; (**f**) Particle size distribution as a function of inlet temperature.

Based on the above results, the mechanism of microsphere formation is proposed and briefly illustrated in Figure 3. First, small droplets form when the emulsion goes through the atomizer. Due to the evaporation of water, the droplets shrink and the latex particles aggregate, leading to the initial formation of the wet shells of the microspheres. Wet shell microspheres are gradually dried and maintain the spherical morphology until the droplets completely solidify. The residual water within the spheres subsequently evaporates through the tiny pores and capillary channels between the latex particles. Finally, the hollow microspheres are generated. According to SEM characterization, this mechanism is applicable to the droplets with a size lower than ca. 20 μm. For larger droplets (>20 μm), there is a large amount of moisture within the wet shell. It evaporates quickly at elevated temperatures (ca. 140–180 °C), which is too much to volatilize through tiny pores or capillary channels, resulting in a gradual increase in internal pressure. When the internal pressure is higher than the mechanical strength of the shell, two situations occur: (a) if the inlet temperature is relatively low (i.e., 180 °C or lower), it creates a drying temperature lower than the glass transition temperature of polyacrylate shell, then the microspheres rapidly burst, generating debris and/or residues; (b) if the inlet temperature is very high (i.e., 200 °C or higher), which generates a surrounding temperature higher than $T_g$ of the shell, then bubble punching would happen due to viscoelastic failure of the shell, instead of breaking the shells.

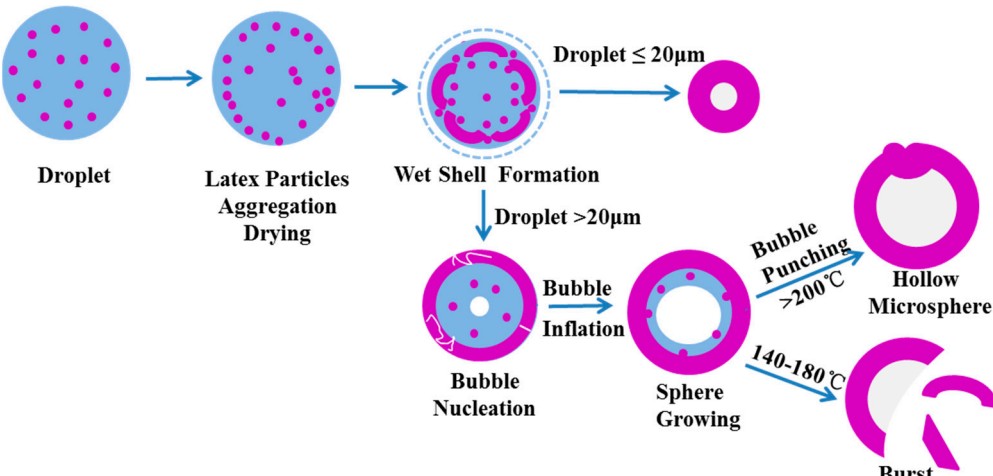

**Figure 3.** Formation mechanism of the polyacrylate hollow microspheres during centrifugal spray drying.

The speed of the atomizer disk directly determines the quality of the atomization effect and size of the atomized droplets, resulting in different microscopic morphologies. Figure 4 shows the effect of atomizer speed on the particle size and size distribution of the formed polyacrylate hollow microspheres at a 220 °C drying temperature and an emulsion feed rate of 160 g/min. With an increasing atomizer speed from 10,000 to 28,000 rpm, the average particle diameter D(50) of the microspheres gradually decreased from 76 to 35 μm. When the speed was further increased to 34,000 rpm, there was no obvious decrease in microsphere size. Overall, the higher the atomizer speed, the larger the centrifugal force and friction force subjected to the emulsion. Therefore, the emulsion is sheared and split into smaller droplets. With an improved atomization effect, the diameter of the dried microspheres is gradually reduced.

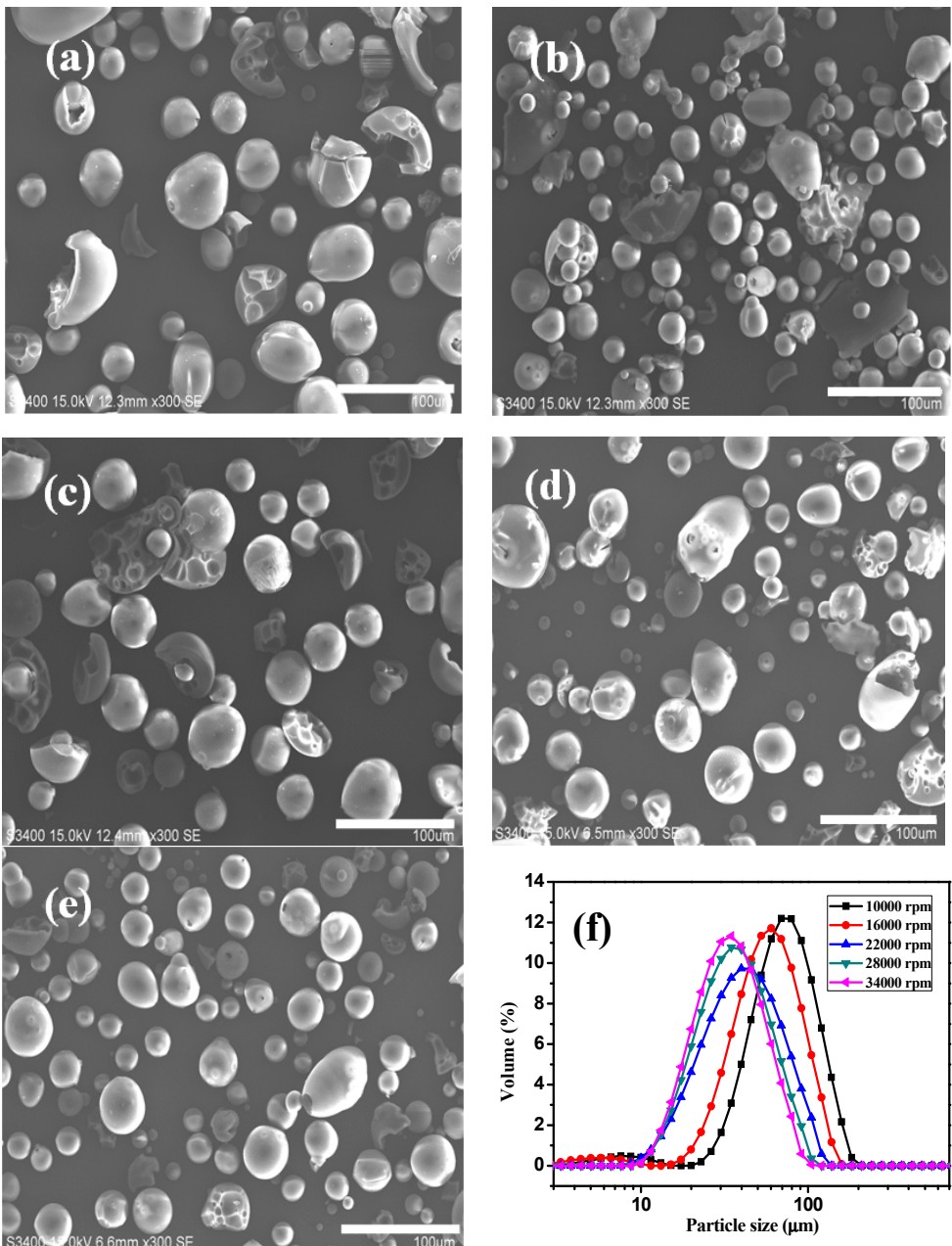

**Figure 4.** SEM images of the microspheres prepared at different atomizer rotational speeds: (**a**) 10,000 r/min; (**b**) 16,000 r/min; (**c**) 22,000 r/min; (**d**) 28,000 r/min, and (**e**) 34,000 r/min; (**f**) Effects of atomizer rotational speed on particle size and size distribution of the formed microspheres at 220 °C drying temperature.

When the inlet temperature of the spray drying (220 °C) and the speed of the atomizer (28,000 rpm) are determined, the drying effect and the size and morphology of the microspheres vary with the change of the feed rate of the emulsion. The effect of feed rate on particle size, size distribution, and morphology of the formed polyacrylate hollow microspheres are shown in Figure 5. At a sufficiently high drying temperature (220 °C), with the feed rate increased from 82 to 190 g/min, there are only marginal changes in the size and shape of the formed microspheres. The microspheres possess a smooth surface, clear interface, and high sphericity. However, some microspheres were broken. Increasing the feed rate can slightly increase the size of the microspheres, and hence the powder fluidity of the microspheres decreases because water evaporation becomes more difficult, but a high productivity can

be obtained. The higher the feed rate, the higher the inlet air temperature required to dry the droplets into spheres. A similar research finding has been reported [23], and if both effectiveness and efficiency are taken into consideration, the optimized feed rate is 136 g/min.

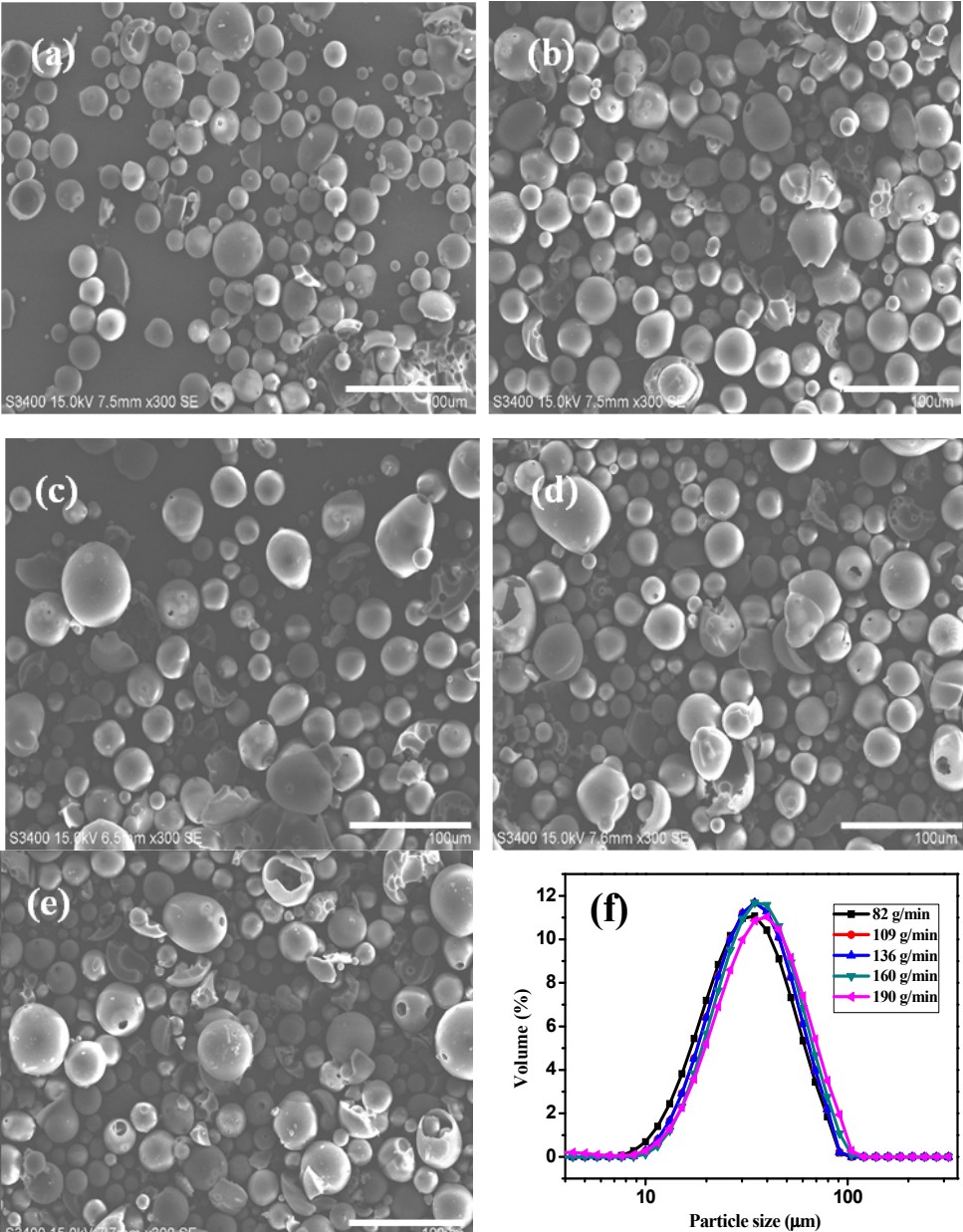

**Figure 5.** Effects of feed rate on particle size, size distribution, and morphology of the formed polyacrylate hollow microspheres: (**a**) 82 g/min; (**b**) 109 g/min; (**c**) 136 g/min; (**d**) 160 g/min, and (**e**) 190 g/min; (**f**) Particle size distribution as a function of feed rate.

## 4. Conclusions

Polyacrylate hollow microspheres were successfully prepared by a facile spray drying method. We found that when the temperature was increased from 140 to 220 °C, the average particle size increased, while the packing density of the microspheres decreased. With an increasing atomizer speed, the average particle size of the microspheres gradually decreased. Feed speed has a marginal effect on the average particle size at a sufficiently high drying capacity, but if the feed rate is too high, the drying process for microspheres will be much harder. We proved that spray drying of polyacrylate emulsion is

a facile and scalable method to prepare polyacrylate hollow microspheres, and this method should be applicable to other systems to prepare various polymer hollow structures for widespread application.

**Author Contributions:** P.C. and N.Y. conceived and designed the experiments; L.S. and Y.L. optimized the process; P.C., N.Y., C.H., L.T., and S.L. performed the experiments; P.C., L.T., S.L., and L.S. wrote the first draft of the manuscript; all authors contributed to revise the manuscript.

**Funding:** This project is sponsored by the National Key Research and Development Program of China (2016YFB0302005), the Science and Technology Program of Guangzhou (201710010176), and the Guangzhou Development District (2017GH32).

**Conflicts of Interest:** The authors declare no conflict of interest.

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
