# Peer review of "Preparation of Polyacrylate Hollow Microspheres via Facile Spray Drying"

_applsci, doi:10.3390/app9020228_

Reviewer 1 Report

This study is interesting and is useful for researchers in the related field. However, some minor revision is required before publication.

1. The intoduction is short. The introduction should be improved

2. In the paper, "The number of damaged microspheres reduced when the inlet air temperature was increased to 220 °C."  However, this is not clear in Figure 2.

Author Response

Reviewer 1:

Question 1: The introduction is short. The introduction should be improved.

Response: Thanks for the comment. We have enriched the introduction with recent developments and applications of hollow microspheres. They have been added to the manuscript.

 Question 2: In the paper, "The number of damaged microspheres reduced when the inlet air temperature was increased to 220 °C."  However, this is not clear in Figure 2.

 Response: Thanks for this comment. As shown in Figure 2(a-c), most of particles show a spherical morphology and a large portion of them are hollow. However, a small amount of broken fragments of microspheres can be observed. Per suggestion by reviewer 1, we removed the comment "The number of damaged microspheres reduced when the inlet air temperature was increased to 220 °C." as it is not obvious in the SEM images.

Reviewer 2 Report

This work presents some systematic attempts to prepare polyacrylate hollow microspheres via spray drying. The work is clear and suitable for publication in Applied Sciences.

 Some minor remarks:

1) Page 2, line 45-48:First, a polyacrylate emulsion with a solid content of 33 wt. % and latex particle size of 30-50 nm was prepared…”. Since the emulsion was synthesized and not purchased, some more details on the reaction and the origin of materials should be given.

2) Page 3, line 82 and also in the Conclusion section: “..microsphere size increased slightly when the inlet temperature increased from 140 to 200 °C.”. This argument is somewhat misleading. From the results of Table 1 and Figure 2f it is clear that the size is rather constant for the temperature range 140-200oC and then slightly increases at 220oC.

3) Do the authors have any information/evidence on the Tg value of the polymer?

Author Response

Reviewer 2:

Question 1:   Page 2, line 45-48: “First, a polyacrylate emulsion with a solid content of 33 wt. % and latexparticle size of 30-50 nm was prepared…”. Since the emulsion was synthesized and not purchased, some more details on the reaction and the origin of materials should be given.

 Response: We have added necessary details in the Experimental section.

 Question 2:  Page 3, line 82 and also in the Conclusion section: “..microsphere size increased slightly when the inlet temperature increased from 140 to 200 °C.”. This argument is somewhat misleading. From the results of Table 1 and Figure 2f it is clear that the size is rather constant for the temperature range 140-200oC and then slightly increases at 220oC.

 Response: Thanks for the comment. We have rephrased the sentences to better describe the phenomenon.

 Question 3:   Do the authors have any information/evidence on the Tg value of the polymer?

Response: Thanks for the comment. The glass transition temperature (Tg) of the synthesized polyacrylate was characterized to be 83 °C by differential scanning calorimetry (DSC) at a heating rate of 10 °C/min. We have included the data in the manuscript.

Reviewer 3 Report

This is an interesting description of a process for making polyacrylate hollow spheres by means of spray drying. I have just a couple of remarks that the authors are kindly requested to take in account for an improved version of their manuscript:

Lines 47-49: There should be more details on the reagents (purity, supplier)

Line 86: The authors should described the criteria for considering a sphere as damaged. Is this only based on SEM images? And if this is the case, how is the borderline defined between damaged and undamaged spheres?

Generally, a more detailed comparison/discussion of the present results with previous work involving different production techniques should be added.

Author Response

Referee 3:

Question 1:  Lines 47-49: There should be more details on the reagents (purity, supplier) 

Response: Thanks for this comment. We have added such details into the manuscript.

 Question 2:   Line 86: The authors should described the criteria for considering a sphere as damaged. Is this only based on SEM images? And if this is the case, how is the borderline defined between damaged and undamaged spheres?

Generally, a more detailed comparison/discussion of the present results with previous work involving different production techniques should be added.

Response: Thanks for this valuable comment. We cited a reference for comparison. The SEM micrographs below show hollow hydroxyapatite microspheres fabricated by the centrifugal spray drying method [1]. One can observe that many microspheres are crushed (Fig. b). However, most of the particles synthesized by our method show a spherical morphology and a large portion of them are hollow, as shown in Figure 2(a-d).

SEM micrographs of the centrifugal spray dried HA microspheres with 7 wt.% NH4HCO3 [1] are provided in the uploaded word file.

Reference:

[1] Jiao, Y.; Lu, Y.P.; Xiao, G.Y.; Xu, W.H.; Zhu, R.F. Preparation and characterization of hollow hydroxyapatite microspheres by the centrifugal spray drying method. Powder Technol., 2012, 217, 581-584.

Round  2

Reviewer 3 Report

I thank the authors for amending their manuscript according to the comments made on the previous version. The manuscript is now acceptable for publication.